# Strain-Balanced InAs/AlSb Type-II Superlattice Structures Growth on GaSb Substrate by Molecular Beam Epitaxy

**DOI:** 10.3390/ma16051968

**Published:** 2023-02-28

**Authors:** Michał Marchewka, Dawid Jarosz, Marta Ruszała, Anna Juś, Piotr Krzemiński, Dariusz Płoch, Kinga Maś, Renata Wojnarowska-Nowak

**Affiliations:** 1Center for Microelectronics and Nanotechnology, Institute of Materials Engineering, University of Rzeszów, Al. Rejtana 16, 35-959 Rzeszów, Poland; 2International Research Centre MagTop, Institute of Physics, Polish Academy of Sciences, al. Lotników 32/46, 02-668 Warsaw, Poland

**Keywords:** molecular beam epitaxy, T2SLS, InAs/AlSb

## Abstract

We demonstrate strain-balanced InAs/AlSb type-II superlattices (T2SL) grown on GaSb substrates employing two kinds of interfaces (IFs): AlAs-like IF and InSb-like IF. The structures are obtained by molecular beam epitaxy (MBE) for effective strain management, simplified growth scheme, improved material crystalline quality, and improved surface quality. The minimal strain T2SL versus GaSb substrate can be achieved by a special shutters sequence during MBE growth that leads to the formation of both interfaces. The obtained minimal mismatches of the lattice constants is smaller than that reported in the literature. The in-plane compressive strain of 60-period InAs/AlSb T2SL 7ML/6ML and 6ML/5ML was completely balanced by the applied IFs, which is confirmed by the HRXRD measurements. The results of the Raman spectroscopy (measured along the direction of growth) and surface analyses (AFM and Nomarski microscopy) of the investigated structures are also presented. Such InAs/AlSb T2SL can be used as material for a detector in the MIR range and, e.g., as a bottom n-contact layer as a relaxation region for a tuned interband cascade infrared photodetector.

## 1. Introduction

The type-II superlattices (T2SL) InAs/AlSb superlattices are very often used as barrier materials for structures of InAs/GaSb applied to infrared photodetectors [1,2]. InAs/AlSb SL can also be used as active materials for a tuned interband cascade infrared photodetector (ICIP) [3]. In the case of the 6.1A family (InAs, GaSb, AlSb) [4], the most important challenge is to obtain the T2SL structures with possible minimum strain versus GaSb or GaAs substrate [5]. Many papers have so far been dedicated to the T2SL InAs/GaSb [6,7,8,9] due to the fact that this SL can be tuned in a wide range of IR radiation from 5 up to 11 μm [10,11]. There has been a significant effort dedicated to improving interface quality in InAs/GaSb T2SL, but relatively little investigation of InAs/AlSb T2SL. From the point of view of the interfaces, in the case of T2SL InAs/AlSb, the situation is similar to that in InAs/GaSb, e.g., [12]. The possible interfaces that can be intentionally added during the MBE process may be of two kinds: InSb (like for both of them, or two different interfaces) and InSb (like on InAs and AlAs, and like on AlSb layers [13]). The theoretical study predicts that the two GaAs-like IFs, in the case of the InAs/GaSb T2SL, cannot balance the strain from mismatch of the lattice constant of the InAs and GaSb layers [14]. These two types (both InSb-like or InSb- and AlAs-like) of interfaces can change the electrical [15] and optical [16] properties of the T2SL InAs/AlSb. This kind of superlattices obtained by the MBE technique have been previously investigated but only on a GaAs substrate [17].

In this paper, we report on the growth of 60-period InAs/AlSb T2SL on GaSb substrate with GaSb buffer layers with different size of the period. A special processes shutter sequence was used in order to obtain the lowest possible stress while maintaining the sharpest possible interface of the two types: InSb and AlAs-like. Such a technique, known as migration enhanced epitaxy (MEE) has so far been applied for InAs/GaAs T2SL [18,19]. The results of HRXRD measurement with the numerical analyses presented here have been completed by measurements of the surface morphology by using a Nomarski optical microscope and an AFM microscope. We also present the room temperature Raman spectroscopy spectra of the presented structures along the direction of growth (z-axis) which confirms the good quality of the investigated structures.

## 2. Experiment

In this work, four samples of InAs/AlSb T2SL based on GaSb substrates with a GaSb buffer layer were investigated. The SLS were grown on GaSb:Te (100) ± 0.5° substrates by using a solid-source Riber Compact 21T (III-V) MBE system, equipped with standard ABN60 dual-zone effusion cells for In, Al, Ga, and with valved arsenic and antimony cracker: VAC 500 and VCOR 300, respectively. An arsenic pyrometer was used to measure the temperature of the substrate.

The substrate rotation speed during growth was set to 20 rpm. The substrate temperature ramp rate was 10 °C/min during the heat-up and 20 °C/min during the cool-down processes. After deoxidation of GaSb, approximately 400 nm GaSb were applied at a temperature of 530 °C to refresh the surface of the substrate. The 60 repetitions of nonintentionally doped InAs (7 ML)/AlSb (6 ML) were carried out at a substrate temperature of 473 °C (sample no. 35D252). Three samples were also prepared of 60 InAs: (GaTe) (6ML)/AlSb (5ML) periods doped with GaTe in the InAs lines (35D257, 35D263, 35D264). In subsequent processes, the GaTe effusion cell temperature were increased by 10 °C, starting from 500 °C.

In Figure 1a, the architectures of the investigated T2SL InAs/AlSb are presented. Figure 1b illustrates interface shutter sequences for InSb-like IF for all samples. The InSb-like IF is formed by deposition: 3 s, 2 s, 3 s, and 2.5 s InSb for samples 35D252, 35D257, 35D263, and 35D264, respectively. After that, the Sb shutter stills open for 4 s, 2 s, and 2.5 s for 35D252, 35D263, and 35D264, respectively. For all samples, the AlAs-like IFs were generated by employing As soak time after AlSb deposition: during 2 s for 35D264 and 35D263 samples and 3 s for 35D257 and 35D252. The thickness of both kinds of IFs were collated in Table 1 after the simulations of the HRXRD measurements. After each InAs and AlSb layer, all shutters were closed for 1 s. The biggest difference from the point of view of the lattice mismatch is observed for two samples with almost the same SLS period: 35D264 and 35D257 (see Table 1). This difference is caused mainly by the different As soak time: 2 s for D35264 and 3 s 35D257. These changes together with the longer time of the Sb soak for the 35D264, caused the mismatch lattice to be reduced to 0.8 × 10${}^{-3}$, which is lower than 0.01%. In this paper, we present results only for four samples for which in our opinion the resuts are most interesting. During the proces of engineering the IFs in the investigated T2SL, many processes were performed. In some of the processes, the different GaTe doped layer was added to InAs layers. The purpose of such procedures is to obtain the proper level of the doping for, e.g., barrier materials such as InAs/AlSb in an application structure in which the InAs/GaSb layers are the absorber layers in the defined IR region [2].

## 3. Results and Discussion

The X-ray diffraction (HRXRD) was carried out by the Malvern Panalytical diffractometer with Cu K${}_{\alpha_{1}}$ radiation (λ = 1.540598 Å). The 2Θ-ω scan was performed around the GaSb (004) reflex. For all samples, the highest intensity and narrowest peak is from the GaSb substrate.

Figure 2 presents the high-resolution X-ray diffraction (red line) pattern around the symmetric 004 reflection and the numerical simulations (black lines) of 60 periods for all investigated T2SL. In all graphs in Figure 2, one diffraction peak originating from the GaSb substrate is exhibited, in addition to well-resolved satellite peaks up to the third order, indicated by “0”, “±1”, etc. This is evidence of the good crystalline quality and the high reproducibility of the InAs/AlSb T2SL. Imperfections in superlattices such as period thickness inhomogeneity or strain relaxation, affect higher-order satellite peaks. Nevertheless, for all measured samples in the 2Θ-ω scan, so-called Pendeläsung fringes can be observed (see Figure 3). They also appear at higher-order satellite peaks, which also proves the high quality of the samples grown [7]. The full-width at half-maximum (FWHM) of the zero-order peak is equal to 54.98, 161.60, 139.78, and 54.62 arcsec for 35D252, 35D257, 35D263, and 35D264, respectively. From the interval between the satellite peaks in Figure 2, the superlattice periodicity of that grown on GaSb substrate was found for all samples, as well as the residual lattice mismatch (Δa/a) (see Table 1). Numerical simulation allowed us to determine the thickness of the InSb-like and AlAs-like interfaces (see Table 1). The low FWHM of the zero-order peak is evidence of the low distributions density in the superlattice layer and demonstrates that both of the interfaces, especially for samples 35D252 and 35D264, have very high structural quality.

Figure 4 shows inverse spatial maps of two superlattices (reciprocal space maps (RSM)) for which the lower mismatch of the lattice constant were obtained. In order to ensure comparability, the same measurement settings were used for each of the structures. Inverse space maps were measured around GaSb 004. As can be seen from the above maps, the inverse lattice points of the substrate, GaSb and “0,” are very close to each other, which is caused by a slight lattice mismatch. Both RSM (a) and (b) show that the reciprocal lattice points corresponding to the ground and SL are perfectly vertically aligned, suggesting that there is no slope in the sample. On both sides of the reciprocal lattice points in the direction of increasing and decreasing Q${}_{z}$ values for Q${}_{x}=$ 0, thickness fringes are visible. The RMS shows the surface streak of the sample (crystal truncation rod (CTR) marked in both Figure 4a,b) and the analizer streak parallel to the diffracted beam (red A in Figure 4a,b). In addition, RSM measurements revealed diffuse scattering (DS) (red circles in Figure 4a,b) around the imaged reciprocal points of the lattice [20]. It is concentrated mainly around turning points 004. It is the largest for sample number 35D252, and the smallest for 35D264 [19].

The surface morphology of the grown InAs/AlSb superlattices (obtained by the Olimpus DSX1000) is shown in Figure 5. As can be seen, all samples exhibit a shiny mirror-like surface. The bigger differences between the samples is caused by the diffrent level of the GaTe dopand (see Experiment Section). The sample 35D252 was undoped, the rest of the samples were doped in different levels. The temperature of the GaTe effusion cell was the highest for the sample 35D264, which means that for this sample, the level of the dopand is the highest. Nevertheless, the best strain balance measured by the mismach of lattice constance is the best for sample 35D264.

The arithmetic mean of the 3D profile elevations (S${}_{a}$) from the 20 μm × 20 μm surface obtained by atomic force microscopy (AFM; Brucker Innova) is 0.24 nm for 35D252 and 35D257, 0.28 nm for 35D263, and 0.31 nm for 35D264 (see Figure 6a–d). These results also show that the undoped sample—35D252—has the lowest value of S${}_{a}$. Nevertheless, all of the samples exhibited excellent morphology with small roughness below 3A. The bigger S${}_{a}$ obtained from AFM measurements for the 35D264 sample is caused by high level of the Te doping. After exceeding a certain value of Te doping (in our research this value was defined at the 5.0 × 10${}^{16}$ cm${}^{-3}$ level) the surface structure strongly degrades.

The Raman spectra were obtained by using an inVia Micro Raman (Renishaw) spectrometer in backscattering configuration and cross-section geometry of samples (the results for InAs/AlSb T2SL samples (35D252, 35D257, 35D263, 35D264) are shown in Figure 7). An argon laser with 488-nm wavelength and 5-mW power was used as an optical excitation source. The Raman scattering was measured by using a Leica DM2500M microscope and objective with 50× lens magnification, yielding laser spot sizes of about 1 μm. Exposure time was set as 5 s with accumulations of 5 scans.

The spectra of InAs/AlSb T2SL structures (black curves in Figure 7a–d) are compared with the spectra of GaSb substrate (red lines of Figure 7a–d). This spectra (black curves) presents two main bands at 227 cm${}^{-1}$ and 236 cm${}^{-1}$, which are associated with the transverse optical (TO) phonons of GaSb and longitudinal optical (LO) phonons of GaSb, respectively. The spectra of InAs/AlSb T2SL are richer in spectral bands. The InAs-TO mode and InAs-LO mode appear at about 220 cm${}^{-1}$ and 235 cm${}^{-1}$, respectively. They partially coincide with the signal from the GaSb buffer. The lines at approximately 317 cm${}^{-1}$ and 340 cm${}^{-1}$ represent AlSb-TO mode and AlSb-LO mode, respectively [21]. The observed slight shifts in the position of the spectral bands between the samples may be related to the occurrence of stresses in the material. The weak line at approximately 189 cm${}^{-1}$ is related to the presence of InSb-like interface. The AlAs-like mode of the second interface is not clearly visible in the registered spectra. The reason for this may be the small thickness of AlAs-like IF compared to InSb-like IF. It contributes to the intense band of InAs LO/GaSb LO and is revealed after the mathematical treatment with the use of the second derivative calculation. The appearance of a bands ranging from 100 cm${}^{-1}$ to 200 cm${}^{-1}$ is considered to be the disorder-activated longitudinal acoustic (DALA) vibrational mode that originates from ionic disorder and interplanar stresses [22]. No distinct bands present in this region could be attributed to well-defined interfaces, and there was no occurrence of interface intermixing.

## 4. Summary

The proper engineering of the IFs in InAs/AlSb T2SL allows us to obtain high-quality superlattices. The formation of these IFs is ensured by As and Sb soaking together with the InSb intentional growth procedure at the IF on InAs. The HRXRD measurements prove the high quality of the investigated structures. Such good quality was obtained by the MEE technique with the special shutter sequences, which allow us to obtain AlAs-like and InSb-like IFs in order to relieve the strain in the superlattices.

The investigated samples during the MBE growth had a different level of the GaTe dopand. The influence of this dopand can be observed in AFM and Nomarski measurements. The best sample—from the point of view of the lattice mismatch—was 35D264, which was doped the most compared to the rest of the samples. The 35D252 sample was undoped, and its surface was the best (see Figure 5a and Figure 6a). During our investigation, we can conclude that the strain balance is a play in which the layer thickness of the SL compared with the proper procedure of the IFs growth plays main roles. The small changes in the opening and closing times of shutters during IF growth cause large changes in lattice mismatch. The additional dopand into the InAs lines in SL caused the morphology of the surfaces to become more distorted, but it does not influence the lattice mismatch.

The results of Raman spectroscopy prove the high quaility of both IFs (weak line at approximately 189 cm${}^{-1}$) is related to the presence of InSb-like interface, and no visible line for AlAs-like IF, which is very thin compared to InSb-like IF.

The developed procedure allows for the scalability of the InAs/AlSb T2SL growth processes for different period sizes, and thus for different ranges of infrared radiation detection.

## Figures and Tables

**Figure 1 materials-16-01968-f001:**
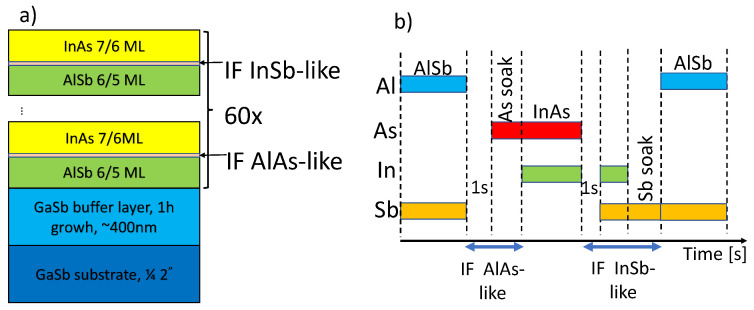
(**a**) The architecture of the InAs/AlSb structures. (**b**) The shutters sequences for the MBE growth of investigated InAs/AlSb T2SL.

**Figure 2 materials-16-01968-f002:**
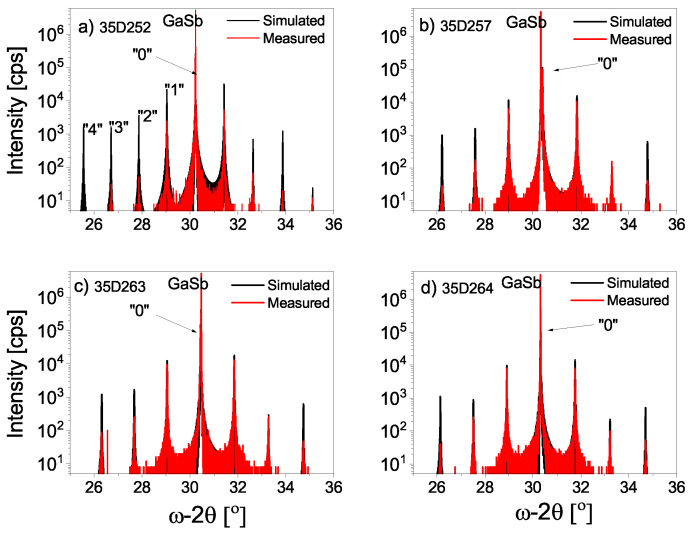
The X-ray 004 2Θ-ω scan of four investigated InAs/AlSb T2SL on GaSb substrate: (**a**) 35D252, (**b**) 35D257, (**c**) 35D263, (**d**) 35D264.

**Figure 3 materials-16-01968-f003:**
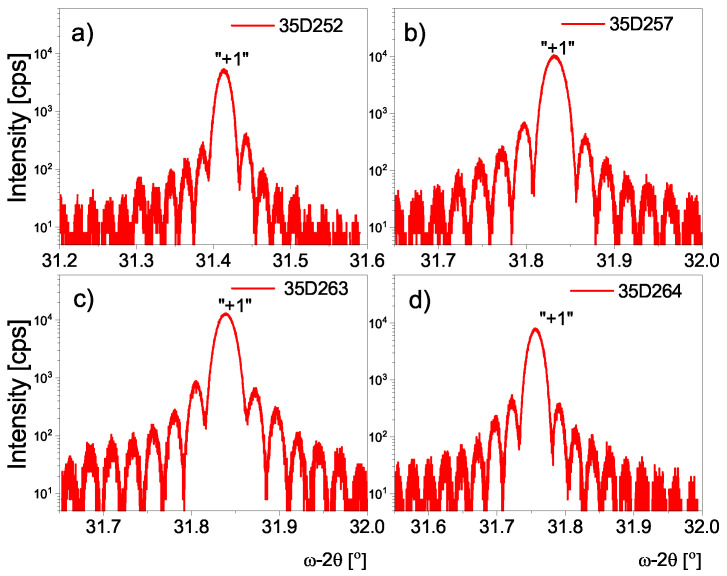
Magnified pictures the first satellite of the HRXRD measurements for all investigated samples: (**a**) 35D252, (**b**) 35D257, (**c**) 35D263, (**d**) 35D264 with the Pendeläsung fringes.

**Figure 4 materials-16-01968-f004:**
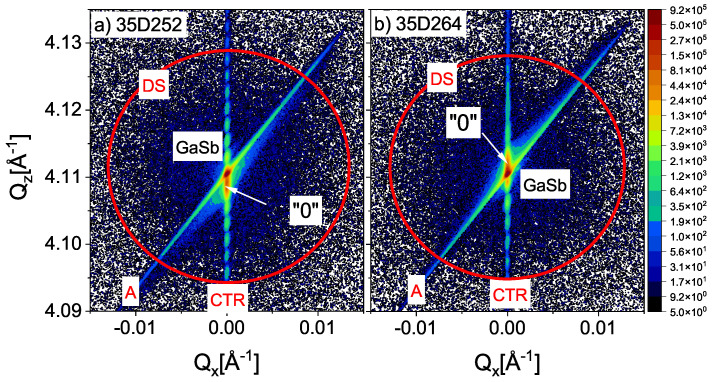
RMS maps for two InAs/AlSb T2SL with the smallest measured lattice mismatch: (**a**) 35D264 (**b**) 35D252. In both panels, crystal truncation rod, CTR; diffuse scattering, DS; red circles; A, analizer streak parallel to the diffracted beam.

**Figure 5 materials-16-01968-f005:**
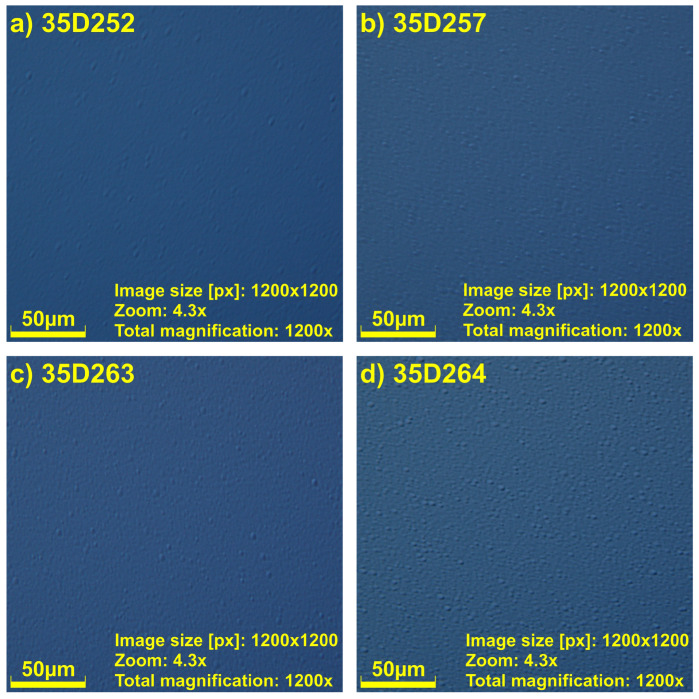
The Nomarski optical microscopy pictures of four InAs/AlSb T2SL samples. (**a**) 35D252, (**b**) 35D257, (**c**) 35D263, (**d**) 35D264.

**Figure 6 materials-16-01968-f006:**
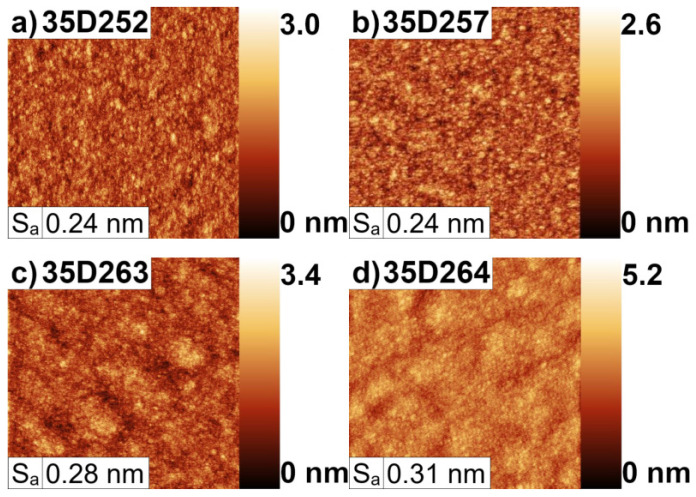
The AFM surface pictures of four InAs/AlSb T2SL samples: (**a**) 35D252, (**b**) 35D257, (**c**) 35D263, (**d**) 35D264.

**Figure 7 materials-16-01968-f007:**
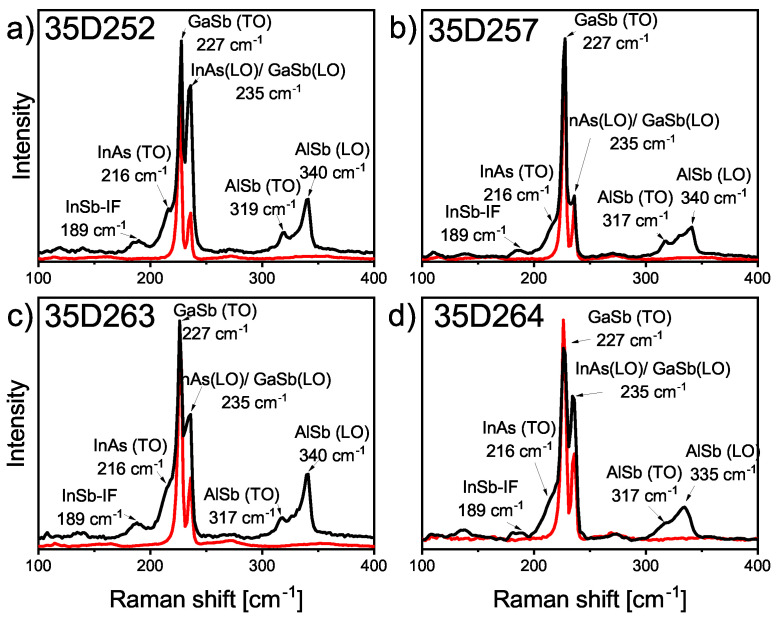
Room temperature Raman scattering spectra of InAs/AlSb T2SL (black curves): (**a**) 35D252, (**b**) 35D257, (**c**) 35D263, (**d**) 35D264 sample against the spectra of GaSb substrate (red lines).

**Table 1 materials-16-01968-t001:** Numerical calculation of the HRXRD measurements.

T2SL	InAs/AlSb	IF${}_{\mathrm{InSb}}$	IF_*AlAs*_	Δa/a
No.	(ML)/(ML)	(ML)	(ML)	× 10${}^{-3}$
35D252	7.1/6.0	0.75	0.42	3.2
35D257	6.0/5.0	0.47	0.43	21.8
35D263	6.0/5.1	0.67	0.35	8.9
35D264	5.8/5.3	0.50	0.35	0.8

## Data Availability

The data that support the findings of this study are available from the corresponding author upon reasonable request.

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
