# Peer review of "Strain-Balanced InAs/AlSb Type-II Superlattice Structures Growth on GaSb Substrate by Molecular Beam Epitaxy"

_materials, 2023, doi:10.3390/ma16051968_

Round 1
Reviewer 1 Report
This is an interesting and well-written paper that seeks to create strain-balanced InAs/AlSb superlattices on GaSb and to understand the effect of the interfacial chemistry. Applications for this work include improved IR photodetectors. The authors should address the following points to strengthen their manuscript.
1. On line 19 the authors say that one goal is to grow 6.1 Å materials with the minimum possible strain on GaSb or GaAs. I understand this with respect to GaSb but I cannot see how these materials can be grown strain balanced on GaAs – the lattice mismatch is too great. They need to clarify this statement.
2. I wonder why the authors decided to call the AlAs type of interface “GaAs-like” instead of just calling it AlAs-like (since no Ga is present in the superlattice).
3. Figure 1 (and some others) would benefit from being larger – at the very least, the font size needs to be larger to make it readable.
4. They do a good job of explaining the differences between the four samples in terms of the SL thicknesses, Te-doping (although they should include the expected carrier density for the two GaTe temperatures used), and interface growth conditions. However, it might be helpful for the authors to explain the overall trend they were seeking to explore across the 4 samples, otherwise it just seems quite random.
5. Given the resolution/step-size of the XRD spectra in Figure 2, it is hard for me to see the Pendellösung fringes that the authors mention in the text. I suggest removing this claim or providing higher resolution spectra that more clearly show these fringes.
6. For Figure 3, the authors should explain why they chose just two of the samples to do RSMs on. If it was too time-consuming to do all four, why did they choose these two specific samples.
7. Figure 3: To help the reader, it would also be helpful to mark on the RSM some of the features they describe in the text (the CTR, DS, monochromator streak etc.)
8. Figure 4: the authors say that there are no defects or hillocks present, whereas I can clearly see “droplet-like” features across the surface of all four samples, although the density varies from sample to sample, with it seems the lowest density for sample 35D252. The authors should comment on the origin of these features and how their size/density correlates with each sample’s growth parameters.
9. In fact, more generally, for each of the characterization techniques, it would be helpful for the authors to comment on how the different results for each sample correlate with their unique structure/growth conditions. For example, in Figure 2/Table 1, 35D264 has the lowest lattice mismatch – why is this? What is it about the authors chosen parameters for that sample that minimized the strain? They should repeat this analysis for Nomarski (see above), AFM, Raman etc.
10. Figure 5 – the authors should define Sa – presumably RMS roughness?
11. Why did the authors choose to dope these structures, especially at such high levels if they observe roughening? Would growing undoped structures have allowed them to better understand the effects of the other variables?
Reviewer 2 Report
Since the 6.1Å family (InAs, GaSb, AlSb) have important application in MIR detector device, the 6.1Å family and its heterostructures have been a popular semiconductor research material. And there's a lot of interesting physics in this 6.1Å family too. The authuors experimently studied strain-balanced InAs/AlSb superlattices grown on GaSb substrates employing two kinds of interfaces (IFs): AlAs-like IF and InSb-like IF. The structures are characterized by the HRXRD measurements, the Raman spectroscopy and AFM and Nomarski microscopy. The material itself is very interesting. However, the author should clarify my following doubts/suggestions:
1) From the perspective of energy band alignment, InAs/AlSb should be a type I quantum well. Why did the author say it is a type II quantum well?
2)When the author reviewed InAs/GaSb quantum wells, they missed the relevant reference: Nature Communications volume 8, Article number: 1971 (2017) .
3)The author studies InAs/AlSb quantum wells. It is understandable that there is an interface similar to InSb, why there is an interface similar to GaAs, should it be an interface similar to AlAs?
4)The authors' physical discussion of the experimental results (Figures 3 to 6) should be strengthened.
Round 2
Reviewer 2 Report
The revised version has incorporated my suggestions and improved a lot. I recommend its publication.
Thanks!